# The Role of Mitochondrial Dysfunction in Atrial Fibrillation: Translation to Druggable Target and Biomarker Discovery

**DOI:** 10.3390/ijms22168463

**Published:** 2021-08-06

**Authors:** Lisa Pool, Leonoor F. J. M. Wijdeveld, Natasja M. S. de Groot, Bianca J. J. M. Brundel

**Affiliations:** 1Department of Physiology, Amsterdam Cardiovascular Sciences, Amsterdam UMC, Vrije Universiteit Amsterdam, 1081 HV Amsterdam, The Netherlands; l.pool2@amsterdamumc.nl (L.P.); l.f.j.m.wijdeveld@student.vu.nl (L.F.J.M.W.); 2Erasmus Medical Center, Department of Cardiology, 3015 GD Rotterdam, The Netherlands; n.m.s.degroot@erasmusmc.nl

**Keywords:** atrial fibrillation, mitochondria, electrophysiology, atrial cardiomyopathy, diagnostics, biomarker, DNA

## Abstract

Atrial fibrillation (AF) is the most prevalent and progressive cardiac arrhythmia worldwide and is associated with serious complications such as heart failure and ischemic stroke. Current treatment modalities attenuate AF symptoms and are only moderately effective in halting the arrhythmia. Therefore, there is an urgent need to dissect molecular mechanisms that drive AF. As AF is characterized by a rapid atrial activation rate, which requires a high energy metabolism, a role of mitochondrial dysfunction in AF pathophysiology is plausible. It is well known that mitochondria play a central role in cardiomyocyte function, as they produce energy to support the mechanical and electrical function of the heart. Details on the molecular mechanisms underlying mitochondrial dysfunction are increasingly being uncovered as a contributing factor in the loss of cardiomyocyte function and AF. Considering the high prevalence of AF, investigating the role of mitochondrial impairment in AF may guide the path towards new therapeutic and diagnostic targets. In this review, the latest evidence on the role of mitochondria dysfunction in AF is presented. We highlight the key modulators of mitochondrial dysfunction that drive AF and discuss whether they represent potential targets for therapeutic interventions and diagnostics in clinical AF.

## 1. Introduction

Atrial fibrillation (AF) is the most prevalent age-related cardiac arrhythmia affecting millions of people worldwide [1]. AF importantly increases the risk of serious heart-related complications such as heart failure and stroke [2,3], and as such, AF contributes substantially to cardiovascular morbidity and mortality [1]. As the population ages globally, AF is expected to affect 6 to 12 million people in the USA by 2050 and 17.9 million in Europe in the next 40 years [4]. AF is characterized by impaired electrical conduction and subsequently contractile dysfunction [5], which is due to specific disruption in the structure of the atrial myocardium [6,7]. This process is defined as electropathology and underlies AF onset and progression [8]. Once initiated, several therapeutic approaches have been developed, including pharmacotherapy [9]. As these treatments are directed at electrical changes and not the underlying structural disruption, they are only moderately effective to attenuate AF [5]. Furthermore, current pharmacotherapy may predispose to considerable risks, especially life-threatening pro-arrhythmia events [9]. Therefore, there is an unmet need for a better understanding of the molecular mechanisms underlying AF, which may aid in the identification of diagnostic biomarkers and effective mechanism-based treatment approaches.

It has been well described that cardiac mitochondria are vital organelles that underlie metabolic processes for the maintenance of cardiac function [10]. Hereto, mitochondria form a dynamic network that represents one-third of the myocardial volume and produces over 95% of the energy [10]. Mitochondria adequately synthesize high-energy molecule adenosine triphosphate (ATP) derived from oxidative phosphorylation, in order to support the mechanical and electrical activities of the heart [11]. As the energy storage within the cardiomyocytes is only sufficient to sustain the heartbeat for a few seconds, a tight coupling of ATP synthesis and myocardial contraction is of paramount importance [12]. For ATP generation, the heart is capable of utilizing various classes of energy substrates, including fatty acids (FAs), glucose, amino acids, and ketone bodies, for oxidative phosphorylation in mitochondria [13,14]. Under normal physiological conditions, cardiomyocytes mainly utilize FAs and carbohydrates as their predominant substrates for oxidative phosphorylation [13]. FA β-oxidation accounts for 60 to 80% of ATP generations, and the remaining ATP is primarily accounted by carbohydrate oxidation [13]. On the flip side, the by-product of mitochondrial bioenergetic activity is the generation of reactive oxygen species (ROS) [15]. Normally, these toxic and damaging by-products are neutralized or scavenged by antioxidants. However, impaired antioxidant defense, due to oxidative stress, increases ROS formation, resulting in profound damaging effects in cardiac cells and as such drives cardiac disease development [16,17,18].

Various studies investigated the role of mitochondrial dysfunction as a contributor to AF development [11,19]. In atrial cardiomyocytes, oxidative stress and mitochondrial DNA damage are increased in patients with AF [20]. This contributes to the impairment of the bioenergetic function of mitochondria and the onset and progression of AF [21,22]. Moreover, a decrease in ATP levels, loss of mitochondrial membrane potential (ΔΨ_mito_) and mitochondrial network fragmentation have been observed resulting in contractile dysfunction and AF progression in experimental and clinical AF [20,23]. These emerging research findings indicate that mitochondrial dysfunction plays a central role in the pathophysiological mechanisms driving AF. The importance of mitochondrial dysfunction in AF and pharmaceutical and nutraceutical compounds directed at mitochondrial dysfunction as a treatment strategy in AF have not been reviewed in detail yet. Especially, the role of the microtubule network to serve mitochondrial function may shed novel insights on mitochondrial dynamics, and may reveal novel treatment options in AF.

In this review, we discuss the current knowledge on the role of mitochondrial dysfunction in AF. Furthermore, we identify possible novel treatment strategies directed at the conservation of mitochondrial function and stabilization of the microtubule network in AF. Finally, we elaborate on key mitochondrial modulators as a potential biomarker in clinical AF.

## 2. Cardiac Mitochondrial Physiology

Given the importance of mitochondria as integrators of cellular energy metabolism and signal transduction, it is foreseeable that mitochondrial dysfunction is associated with various human diseases, including AF. Therefore, conservation of mitochondrial function may represent a crucial step in the therapy of this cardiac arrhythmia. To understand the role of mitochondrial dysfunction in AF, the physiological processes are first elucidated in order to move towards their role in pathophysiology and mitochondria-targeted therapies.

### 2.1. ATP Synthesis by Fatty Acid and Carbohydrate Oxidation

To understand how mitochondrial dysfunction drives AF, it is important to understand the physiological role of mitochondrial processes in healthy atrial cardiomyocytes. The high cardiac demand for ATP is predominantly recharged by mitochondrial oxidative phosphorylation. Under physiological conditions, FA β-oxidation accounts for 60 to 80% ATP generation [13,14]. For the FA β-oxidation pathway, medium-chain FAs pass the mitochondrial membranes without prior activation of acyl Co-enzyme A (acyl-CoA) [24]. In contrast, the mitochondrial membrane is impermeable to long-chain FAs. Therefore, long-chain FAs need to be activated to form acyl-CoA followed by entry into the mitochondria via the carnitine acyltransferase membrane shuttles, including carnitine palmitoyltransferase I and II (CPT-I and CPT-2) and carnitine-acylcarnitine translocase (CACT) [25]. As depicted in Figure 1, acyl CoA is converted into acylcarnitine by the action of CPT-1 complex. Subsequently, acylcarnitine is further translocated across the inner mitochondrial membrane (IMM) by CACT. In the mitochondria, CPT-2 converts acylcarnitine back into acyl-CoA, the substrate for mitochondrial FA β-oxidation [26].

During FA β-oxidation, 2-carbon units are oxidized of the fatty acyl-CoA molecule to produce acetyl Co-enzyme A (acetyl-CoA) [24]. Acetyl-CoA enters the citric acid cycle (TCA cycle), where it is further oxidized into CO_2_ with the concomitant generation of nicotinamide adenine dinucleotide (NADH) and flavin adenine dinucleotide (FADH_2_) [27]. Electrons from NADH and FADH_2_ are donated to the electron transport chain (ETC) at either complex I or complex II, respectively. The electrons from complexes I and II are donated to ubiquinone and reduced to ubiquinol. Ubiquinol is oxidized in complex III, allowing for the transport of electrons to cytochrome C. The transfer of electrons induces the pumping of protons into the intermembrane space. Cytochrome C transports electrons to complex IV, where oxygen is reduced to water. This reduction results in the pumping of protons to the intermembrane space. This translocation of protons from the mitochondrial matrix into the intermembrane space in response to electron transfer, establishes a proton gradient. Subsequently, protons are pumped back into the matrix through F0F1-ATP-synthase (complex V), which is coupled with the generation of adenosine triphosphate (ATP) from phosphorylated adenosine diphosphate (ADP) [28,29].

During abrupt increases in workload, the relative contribution of glucose utilization for ATP production increases. Glucose is converted into pyruvate via the glycolytic pathway, which in turn, is a substrate for ATP synthesis (Figure 1) [30]. In contrast to the complex traverse of FAs, pyruvate traverses the mitochondrial membrane through a voltage-dependent anion channel (VDAC) and subsequently crosses the IMM by the mitochondrial pyruvate carrier (MPC) [31]. Once in the mitochondrial matrix, pyruvate can be oxidized into acetyl-CoA by the pyruvate dehydrogenase complex (PDC). Acetyl-CoA enters the TCA cycle to produce the reducing equivalents NADH and FADH_2_ to generate a proton gradient required for oxidative phosphorylation and ATP synthesis [29]. Thus, cardiomyocytes rely on ATP synthesis in the mitochondria. FAs are the predominant substrate for cardiac energy production. During an abrupt increase in workload, mitochondria increase the synthesis of ATP to meet the higher energetic requirements of the heart. However, over time, alternations in mitochondrial function reduce energetic reserves and deteriorate cardiac electrical and mechanical functions, contributing to the development of the substrate for AF [23,32].

### 2.2. Reactive Oxygen Species Generation during Oxidative Phosphorylation

Apart from energy production, ROS are formed as by-products of mitochondrial bioenergetic activity [15]. Not all of the electrons derived from substrate oxidation are transferred to the final electron acceptor in the mitochondrial ETC. Instead, these electrons directly leak out of the ETC into the matrix of the mitochondria. Here, electrons interact with oxygen to generate superoxide (Figure 1) [15]. The concentration of ROS determine whether ROS activate physiological or pathophysiological pathways. Therefore, this concentration is a delicate balance between the rate of ROS production and the rate of clearance by scavenging mechanisms, such as the mitochondrial antioxidant defense systems [19,33]. Tissue or cells are in a healthy state if the rate of production and scavenging capacity is essentially constant and balanced [34]. A disturbed ROS balance, either by ROS overproduction or decreased scavenging activity of the antioxidant systems, contributes to a broad spectrum of human diseases, including cancer, mitochondrial disease, diabetes, aging, and AF [6,35,36]. Oxidative stress and mitochondrial dysfunction induce DNA damage, genome instability, and changes in metabolic signaling pathways, which contribute to the pathogenesis of AF [6,20]. However, the role of mitochondrial oxidative stress in AF is already reviewed in detail [16,37,38,39,40]. Therefore, we keep this topic out of consideration. On the flip side, if the initiated increase in ROS is relatively small, the antioxidative response may be sufficient to scavenge the increase in ROS and resets the original balance [34].

### 2.3. Key Role of SR-Mitochondrial Ca^2+^ Handling in Atrial Cardiomyocyte Contractile Function

Accumulating evidence points to the involvement of mitochondria in Ca^2+^ homeostasis and its Ca^2+^ buffering capacities, for cardiac excitation-contraction (E-C) coupling [41,42]. E-C coupling is the process that links electrical activation to cardiac contraction. Ca^2+^ handling is therefore an essential physiological process for cardiac contraction and function. E-C coupling is initiated by the opening of L-type Ca^2+^ channels (LTCC). Subsequently, Ca^2+^ entry prompts for larger and sustained Ca^2+^ release from the sarcoplasmic reticulum (SR)-Ca^2+^ stores via type 2 ryanodine receptor (RyR2) channels by a mechanism known as Ca^2+^ induced Ca^2+^ release (CICR) which triggers the contraction of the cardiomyocyte [42,43]. Cytosolic Ca^2+^ binds to cardiac troponin-C on the myofilaments, which moves the troponin complex away from the actin binding site [41]. The removal of the troponin complex facilitates the binding of myosin to actin. The myosin head binds to ATP and pulls the actin filaments toward the center of the sarcomere, initiating cardiac contraction [41]. Followed by contractions, the RyR2 and L-type Ca^2+^ channels usually remain closed. In addition, cytosolic Ca^2+^ crosses the outer mitochondrial membrane (OMM), through voltage-dependent anion channels (VDACs) and is transported into the mitochondrial matrix via the mitochondrial Ca^2+^ unit (MCU) (Figure 2) [44]. The efflux of Ca^2+^ from the mitochondria matrix is mediated by the mitochondrial Na^+^/Ca^2+^ exchanger (NCLX). Excessive levels of cytosolic Ca^2+^ are removed mainly by returning Ca^2+^ back to the SR via Ca^2+^ ATPase (SERCA) pump [45,46]. Dropping of the intracellular Ca^2+^ concentration, returns the troponin complex to the resting position, thereby effectively inhibiting cardiac contraction and initiating relaxation. During a contraction-relaxation cycle, mitochondrial Ca^2+^ uptake and buffering have the potential to shape Ca^2+^ transients and therefore contribute to the regulation of contractile function [47]. Intriguingly, the mitochondria are tightly coupled to the SR via the microtubule network and hereby mediate intracellular Ca^2+^ handling during every heartbeat [48]. A balanced Ca^2+^ entry into the mitochondria is required, as mitochondrial Ca^2+^ overload can result in increased mitochondrial membrane permeability, mitochondrial damage, reduced ATP production, and oxidative stress [49]. Together, Ca^2+^ transportation from the SR to the mitochondria via tight coupling of the microtubule network regulates cardiac contraction and preserves mitochondrial function.

### 2.4. Interactions between Sarcoplasmic Reticulum and Mitochondria Safeguards Cardiac Function

It has been recently established that the mitochondria and the SR are well connected by the microtubule network [41,50,51,52]. In cardiomyocytes, the stability of the microtubules affects both mitochondrial function and Ca^2+^ homeostasis, both essential to preserve cardiac contractility and function. The function of the microtubule network, which consists of α and β tubulin, is mainly regulated by posttranslational modifications of the tubulin subunits [53]. Accordingly, acetylated α-tubulin is implicated in regulating the stability and function of the microtubule network. Tubulin interacts directly with the VDACs that are located on the OMM [54], emphasizing a key role for VDACs in the control of effective signal transduction between the two organelles, and mitochondrial function and cardiac contraction [55,56]. The VDAC forms channels on the OMM for mitochondrial membrane permeability and the flux of metabolites into the mitochondria for oxidative phosphorylation. VDAC channel activity and subsequent ΔΨ_mito_ is modulated by the flux of respiratory substrates, including NADH [57], ATP [58], and glutamate [59]. Altogether, the mitochondrial function is strongly related to the microtubule network and respiratory substrates that are controlled by VDAC status and OMM permeability. This interaction safeguards mitochondrial energetics regulation and cell function [54].

In addition to the microtubule network, SR and mitochondria are physically linked via mitofusin 2 (Mfn2) tethers, which enables Ca^2+^ crosstalk between the two organelles via VDAC (Figure 2) [60,61,62]. Hereto, VDAC contains Ca^2+^ bindings sites and is therefore highly permeable for Ca^2+^ and as such modulates Ca^2+^ levels in the mitochondria intermembrane, followed by transportation to the mitochondrial matrix via MCU [44]. Once inside the mitochondria, Ca^2+^ stimulates the Krebs cycle activity and ATP synthesis for cardiac contraction [63]. The subsequent increase in Krebs cycle products, NADH and FADH_2_, forms the substrates of both the ETC and nicotinamide nucleotide transhydrogenase (NNT) [64]. The NNT complex sustains mitochondrial antioxidant capacity through the generation of nicotinamide adenine dinucleotide phosphate (NADPH) [56,65]. However, the mitochondrial import of SR-derived Ca^2+^ is contingent upon the proper physical distance between these two organelles [66]. Loss of contact between the SR and mitochondria upon reduced Mfn2 disrupt mitochondrial Ca^2+^ transfer and contributes to myocardial oxidative stress [62,67]. Considering the importance of the microtubule network for Ca^2+^ transient handling, it is not surprising that disruption of the microtubule network or Mfn2 impairs Ca^2+^ handling, contractility, and mitochondrial function [68,69]. Conservation of the physical connection between both organelles is therefore of utmost importance to ensure atrial cardiomyocyte function.

### 2.5. Mitochondrial Heat Shock Proteins Critical for Mitochondrial Gene Expression and Function

Mitochondria contain their DNA in the form of mitochondrial DNA (mtDNA) [70]. In humans, the mtDNA encodes for 13 mitochondrial proteins. These mitochondrial encoded proteins are key parts of the oxidative phosphorylation complexes to produce ATP. Hence, proteins encoded by the mitochondria are essential for mitochondrial function and cardiac contraction [71]. Nonetheless, the majority of the mitochondrial proteins are encoded by the nucleus [70,72]. These proteins are synthesized in the cytosol and subsequently, imported into the mitochondria. The nucleus controls mitochondrial activity by modulating the expression of nuclear-encoded mitochondrial proteins that regulate mitochondrial gene expression, proteostasis, protein import, oxidative phosphorylation, and metabolism [70,72]. Accordingly, the mitochondrial gene expression system is engaged in an inter-compartmental crosstalk with the cell nucleus to ensure proper mitochondrial function and, accordingly, cardiomyocyte function. In the reverse direction, mitochondrial dysfunction occurs via mislocalization or misfolding of mitochondrial proteins, low levels of ATP, and elevated levels of ROS due to oxidative phosphorylation defects [71]. The transcription of the nuclear-encoded mitochondrial heat shock proteins (HSPs), including HSP60 and HSP10, are essential proteins to maintain mitochondrial protein homeostasis and function. HSP60 plays a pivotal role in the regulation of protein folding and the prevention of protein aggregation in the mitochondria [73]. The mitochondrial chaperonin HSP10 is involved in the folding and assembly of proteins imported into the mitochondrial matrix [74]. High levels of HSP60 and HSP10 in cardiomyocytes have protective effects on the complexes of the ETC. Especially, complex III and IV are significantly upregulated in HSP60- or combined HSP60/10-overexpressed cardiomyocytes [75]. Whereas, deletion of HSP60 in adult cardiomyocytes results in diminished mitochondrial complex activity, reduced ΔΨ_mito_, and increased ROS production in *mice* [76]. These findings indicate that transcription of nuclear-encoded mitochondria such as HSPs is required for normal cardiac function by preserving mitochondrial function.

## 3. The Role of Mitochondrial Dysfunction in the Pathophysiology of AF

In recent decades, the pathophysiology of AF has been studied intensively to understand its underlying initiation and perpetuation mechanisms. Although the exact role of mitochondria in the pathogenesis of AF is not completely elucidated, it is known that mitochondria contribute to the electropathology of this cardiac arrhythmia. An overview of research findings in experimental and clinical AF is given in Table 1.

### 3.1. Mitochondrial NAD^+^ Depletion Confers Cardiomyocyte Dysfunction in AF

A recent study identified a key role of mitochondrial NAD^+^ depletion in oxidative DNA damage and the pathophysiology of AF [6]. Oxidative DNA damage, caused by oxidative stress, is characterized by the accumulation of mitochondrial ROS and insufficient ability of the antioxidant defense mechanisms to detoxify these free radicals [19,33,80]. NADPH oxidase emerges as the primary initiating source for cardiac ROS production and the promotion of AF [81,82,83], which is reviewed by Ren et al. [84]. Likewise, Kim et al. showed that NADPH-driven ROS production, reflective of NADPH activity, was significantly increased in the right atrial appendage in patients with postoperative AF [85]. Interestingly, NADPH-driven ROS production underlies the onset and maintenance of electrical remodeling in AF, which can be successfully prevented with a novel gene-based approach which inhibits NADPH-driven ROS production in AF models [86]. Generation of ROS levels that exceed the detoxification capacity of the cell leads to impaired mitochondrial function through oxidative damage of DNA, proteins, and lipids [87,88]. In response, DNA repair pathways are activated to overcome this oxidative DNA damage. Poly-ADP ribose polymerase (PARP), especially PARP1, is one of the six nuclear enzymes that recruit the DNA repair machinery [6,89,90]. In turn, PARP consumes NAD^+^ in order to synthesis PAR chains. Progressive and excessive PARP activation results in NAD^+^ depletion to an extent that it depletes mitochondrial energy substrate. Therefore, oxidative stress induces mitochondrial dysfunction by promoting the rapid loss of ΔΨ_mito_ and attenuating mitochondrial energy production [6,91,92,93]. Whereas, the inhibition of PARP1 and energy repletion preserves ΔΨ_mito_, NAD^+^ levels, and mitochondrial DNA damage after oxidative stress [91]. Taken together, cardiomyocyte dysfunction during AF may be a consequence of oxidative stress, which consequently induces mitochondrial NAD^+^ depletion and exacerbates mitochondrial dysfunction, structural damage, DNA damage, and electrophysiological deterioration. These findings point to a prominent role of mitochondrial NAD^+^ in the electropathology of AF.

### 3.2. Alternations in the Microtubule Network Contribute to AF

Since the SR and mitochondria are closely connected, it is not surprising that disruption of the microtubule network impairs mitochondrial and cardiac function. Nowadays, it has been recognized that loss of contact between the organelles results in Ca^2+^ overload within mitochondria and eventually leads to mitochondrial and contractile dysfunction [49], which is reviewed by Ruiz-Meane et al. [94]. Particularly, emerging evidence points to a key role in microtubule disruption and AF [7,77,79]. Trafficking of metabolites through the cell as well as contractile function is mediated by acetylated α-tubulin [7]. Histone deacetylase 6 (HDAC6) functions as a potent α-tubulin deacetylase, and thereby regulates the microtubule network. In cardiomyocytes, overexpression of HDAC6 results in the deacetylation and degradation of the microtubules [95]. On the other side, inhibition of HDAC6 increases microtubule acetylation levels and thereby conserves the microtubule structure [7,96]. In line with experimental data, patients with persistent AF reveal increased HDAC6 activity and increased deacetylation and degradation of α-tubulin levels in the atria. In addition, HDAC6-induced deacetylation and degradation of the microtubule network may underlie mitochondrial dysfunction, as observed in experimental and clinical AF [7]. Altogether, disruption of the microtubule network, as observed in AF, may play a role as a key modulator causing mitochondrial dysfunction and disease progression.

## 4. Potential Mitochondrial Markers for AF Diagnostics

So far, no specific biomarkers have been identified to determine the stage of AF severity. There are indications that modulators of mitochondrial dysfunction may aid in the staging of AF and predict the outcome of AF treatments. These modulators include 8-hydroxy2′-deoxyguanosine (8-OHdG), circulating cell-free mitochondrial DNA (cfc-mtDNA), and HSPs (see Figure 3). These markers may be potential biomarkers in the clinic for AF diagnosis, as current biomarkers are not directed at the conservation of mitochondrial function and no biomarkers are available to stage the severity of AF. Therefore, mechanism-based biomarkers may potentially overcome this shortcoming.

### 4.1. Oxidative Stress Marker 8-hydroxy-2′-deoxyguanosine in AF

Cardiomyocytes of patients diagnosed with persistent AF show significant oxidative protein and DNA damage in atrial tissue due to NAD^+^ depletion [6]. In nuclear and mitochondrial DNA, 8-OHdG is one of the predominant forms of free radical-induced oxidative lesions and has therefore been widely used as a serum biomarker for oxidative stress in diseases [97]. As an increase in oxidative protein and DNA levels have been observed in human atrial tissue samples of patients with AF, 8-OHdG may represent a serum biomarker for AF [98]. A recent study revealed a positive correlation between 8-OHdG levels and the stage of AF. Serum 8-OHdG levels were increased in paroxysmal and persistent AF patients compared to control sinus rhythm patients [20,98]. 8-OHdG can even distinguish the stage of AF, as the level gradually increases during more advanced stages of AF [98]. Moreover, patients who developed POAF showed elevated levels of 8-OHdG compared to patients without POAF [98]. This observation is in line with findings that POAF is mainly induced by the production of ROS, primarily via enhanced NADPH oxidase activity due to mitochondrial dysfunction [99]. Comparable results were also found in the study of Toyama et al. where 8-OHdG levels were significantly increased in AF patients compared to sinus rhythm patients, and reduced again after ablative therapy [100]. Thus, the findings indicate the potential diagnostic value of 8-OHdG as a biomarker in both, staging of the severity of AF as well as predicting POAF onset after cardiac surgery. This approach, that key modulators of molecular pathways may act as blood-based biomarkers, seems promising to identify mechanism-based biomarkers to stage the severity of AF. The origin of the biomarkers has not been elucidated yet.

### 4.2. Circulating Cell-Free Mitochondrial DNA in AF

Cfc-mtDNA is successfully used as a biomarker for conditions associated with mitochondrial diseases, since mitochondrial stress and damage trigger the release of cfc-mtDNA in the circulation [101]. As AF is associated with mitochondrial dysfunction and damage [23], cfc-mtDNA may serve as a potential biomarker to predict the incidence and progression of AF. Recently, the levels of cfc-mtDNA were measured in blood samples of patients with and without AF [101]. Cfc-mtDNA in the blood was increased in early paroxysmal AF, especially in man, compared to control samples in sinus rhythm and longstanding persistent AF. This makes cfc-mtDNA a potential biomarker to identify patients with early-stage paroxysmal AF. Currently, the origin of cfc-mtDNA in the serum is unknown. Interestingly, the presence of mitochondrial DNA in the medium of tachypaced atrial cardiomyocytes confirms the release of mtDNA into the circulation [101]. This indicates that cfc-mtDNA in serum of AF patients may originate from atrial cardiomyocytes. As more severe stages of AF are associated with mitochondrial damage and dysfunction [101], the release of mtDNA into the circulation may represent the early disease process that becomes exhausted during more persistent AF stages [101]. Furthermore, in patients, increased cfc-mtDNA levels were positively associated with AF recurrence after treatment [101]. Together, these findings indicate that AF drives the release of mitochondrial DNA from the cardiomyocytes into the circulation, thereby representing a potential biomarker for the stage of AF and predict recurrence after AF treatment.

### 4.3. Mitochondrial Heat Shock Proteins in AF

Currently, there has been increasing interest in nuclear-encoded mitochondrial HSPs as a potential serum biomarker in AF [102]. Interestingly, HSPs have been shown to halt or even reverse structural damage in experimental AF model systems [103]. Moreover, increased levels of mitochondrial HSPs, including HSP60 and HSP10, were observed in the atrial myocardium of patients with severe AF compared to sinus rhythm patients which may reflect higher energy metabolism [104,105,106]. However, several publications found no association between baseline mitochondrial HSP levels and the presence of AF [101,107,108]. In addition, no link was found between mitochondrial HSP levels and AF stages and recurrence after therapy [107]. These findings imply that, in contrast to mtDNA, nuclear-encoded mitochondrial HSPs may not be an applicable blood-based biomarker for AF.

## 5. Mitochondria as a Target for Therapeutic Interventions

As previously described, considerable progress has been made in unraveling the underlying molecular mechanisms as to though which mitochondria drive AF. Consequently, several new pharmaceutical and nutraceutical compounds are under investigation, these include compounds directed at the conservation of mitochondrial function and stabilization of the microtubule network, see Figure 3.

### 5.1. Pharmaceuticals to Conserve Mitochondrial Function

The two compounds, Ru360 and SS31, target mitochondrial mechanisms via inhibition of the MCU and conservation of the mitochondrial ETC complexes and bioenergetics, respectively [109]. The ability of SS31 and Ru360 to conserve mitochondrial function has been tested in AF. In experimental model systems of AF, Ru360 was found to prevent mitochondrial Ca^2+^ overload, and downstream mitochondrial and consequently contractile dysfunction [23,110,111]. So far, Ru360 is only used in pre-clinical settings. The antioxidant SS31 is currently tested in clinical trials. SS31 improves the coupling of the ETC complexes and thereby enhances the mitochondrial bioenergetics and suppresses ROS abundance and oxidative stress [112]. In experimental AF models, SS31 protects against the loss of mitochondrial respiration by increasing ATP production after tachypacing [23]. In addition, SS31 has been shown to protect against network fragmentation [23]. These results reveal a protective role of SS31 against mitochondrial dysfunction and stress in experimental AF settings. Together, targeting mitochondria by SS31 or Ru360 may present novel therapeutic strategies to counteract AF-induced mitochondrial dysfunction.

In addition to the direct targeting of mitochondria, conservation of the microtubule network may also aid in mitochondrial function. Acetylation of α-tubulin has been implicated in regulating microtubule stability and function. Deacetylation of a microtubule network by HDAC6 results in disruption of the network and underlies AF potentially via uncoupling of the SR and mitochondria [113]. The study of Zhang et al. showed that HDAC6 inhibition by tubacin prevents de-acetylation of α -tubulin and thereby suppresses its degradation by calpain [7]. Consequently, the atrial cardiomyocytes were protected against AF in cultured atrial cardiomyocytes, *Drosophila*, and *canine* models for AF [7]. Whether the protective effect is via the prevention of mitochondrial dysfunction is unknown. As several HSPs such as HSP27 were also found to conserve the microtubule network in atrial cardiomyocytes [114], HSP-inducing compounds may have a beneficial effect in AF. A well-known HSP-inducing compound is GGA. The protective effects of GGA in several experimental AF models have been shown [114,115,116]. Interestingly, GGA has the potential to restore the microtubule network and α-tubulin levels after a period of rapid pacing, which coincides with repressed HDAC6 activity [77,117]. These findings pave the way for further studies on the role of HSP induction by compounds, such as GGA, in stabilizing the microtubule network and to protect against AF. Hence, these data indicate the HDAC6 inhibitors and HSP inducers may conserve the microtubule network resulting in conserved mitochondrial function and prevention of AF.

### 5.2. Nutraceuticals to Conserve Mitochondrial Function

Mitochondrial dysfunction has been clearly indicated in AF. Given that the reducing equivalents of NADH and FADH_2_ drive the ETC, the equilibrium of NADH/NAD^+^ is crucial for mitochondrial oxidative phosphorylation. Therefore, NAD^+^ plays a central role in oxidative phosphorylation, making it an attractive target for interventions in cardiac diseases including AF. Indeed, accumulating studies have already shown that boosting NAD^+^ levels improve mitochondrial function in cardiovascular diseases, such as heart failure and ischemic conditions [118,119,120], which has been reviewed in-depth [121]. Collectively, these findings suggest that NAD^+^ homeostasis improves mitochondrial function by attenuation of oxidative stress and DNA damage [118]. Synthesis of NAD^+^ can be initiated from three NAD^+^ precursors, nicotinic acid (NA), nicotinamide, or nicotinamide riboside (NR) [122]. Nicotinamide and NR are converted to nicotinamide mononucleotide (NMN) in the salvage pathway and NAD^+^ is synthesized from nicotinic acid via the Preiss-Handler pathways [122]. The ability of the three precursors to stimulate NAD^+^ biosynthesis varies. However, recent animal studies have indicated that oral supplementation of NR is superior to NA and nicotinamide in elevating NAD^+^ [123]. Recently, the reduction in NAD^+^ levels due to NAD^+^-consuming enzyme PARP has been recognized as a contributing factor to mitochondrial dysfunction and DNA damage in AF [6]. While the association between NAD^+^ and mitochondrial function has been explored in a variety of heart conditions, further research is required to reveal the interaction between NAD^+^ supplementation and AF. The newly elucidated role of NAD^+^ depletion opens the opportunity for NR as a potential novel therapeutic strategy in clinical AF.

Next to NAD^+^ supplementation, mounting evidence has shown growing appreciation for the role of L-glutamine in cardiac dysfunction. L-glutamine is an important mitochondrial substrate that is implicated in the protection of cardiomyocytes from oxidative stress. The TCA cycle can be fed by the substrate L-glutamine, which enhances the production of the reducing agent NADH and FADH_2_ for oxidative phosphorylation [124,125]. Nowadays, L-glutamine is seen as a potential nutraceutical for AF treatment, as it stabilizes the microtubule network by enhancing HSP expression in degenerative and inflammatory diseases [126,127]. Moreover, L-glutamine contributes to the suppression of ROS and ROS-Induced DNA damage due to their antioxidant activity [128,129]. Therefore, L-glutamine metabolisms play an important role in genomic integrity by stabilizing the microtubule structure and suppressing ROS accumulation. The observed effect of L-glutamine on DNA damage, mitochondrial bioenergetics, and microtubule network reveals the importance of L-glutamine as a nutritional therapy for AF onset and progression. Recent data showed for the first time the effect of L-glutamine on metabolites and serum HSP levels in patients with AF [130]. HSP27 and HSP70 were significantly decreased after 3 months of L-glutamine supplementation complemented by the normalization of metabolic pathways [130]. This knowledge may fuel future clinical studies on L-glutamine to improve cardiac function that may attenuate AF episodes.

Finally, CoQ10 may represent another therapeutic option in the treatment of AF patients. Preclinical studies indicate that CoQ10 plays a prominent role in mitochondrial oxidative phosphorylation by facilitating ATP production. Within the ETC, CoQ10 accepts electrons from complex I and II and transports them to complex III. Besides its critical role in the ETC, CoQ10 acts as a potential anti-inflammatory agent [131]. Preclinical data has already provided valuable information that supports the pathophysiological role of CoQ10 in patients with heart failure and ischemic heart disease. Lower CoQ10 levels are seen in patients with ischemic heart disease and advanced heart failure symptoms [132]. The circulating levels of CoQ10 decrease in proportion to the severity of heart injury [133]. In contrast, CoQ10 supplementation results in a significant increase in both serum and myocardial CoQ10 levels [134]. This provides a protective effect on oxidative stress biomarkers and genomic stability [132]. To date, only a few studies have been performed on humans. So far, human studies showed that Q10 supplementation significantly reduces DNA damage markers in serum [135,136]. In addition, CoQ10 treatments have shown a beneficial effect in terms of reducing the incidence of AF in humans [137]. With regard to the published literature on CoQ10 and its beneficial effect on mitochondrial function and suppression of DNA damage, there is a rationale for examining CoQ10 as a possible treatment for AF.

## 6. Clinical and Future Perspectives

In this review article, we present novel findings showing a key role for mitochondrial dysfunction in the onset and progression of AF. Current therapeutic strategies for AF are aimed directly at the suppression of AF symptoms but are not effective in terms of preventing AF progression. Therefore, novel strategies that conserve mitochondrial function may have beneficial effects in AF. Nowadays, several compounds are studied in clinical settings, such as SS31, L-glutamine, NR, and Co-Q10. With regards to the published literature and its beneficial effect on mitochondrial function, there is further rationale for examining CoQ10, L-glutamine, and NR as future treatments in AF. Especially, the newly elucidated role of NAD^+^ depletion in AF provides the opportunity for NR as a potential novel therapeutic strategy in clinical AF [6].

Furthermore, no AF-specific biomarkers are available. So far, several mitochondrial biomarkers have been tested in clinical AF [98,101]. Recent findings indicate the potential diagnostic value of blood-based 8-OHdG and cfc-mtDNA in staging AF. However, clinical studies targeting mitochondrial biomarkers are limited. Additional research is warranted to conclusively prove their value as a biomarker to diagnose clinical AF.

Finally, future research should elucidate whether the described treatments will benefit all AF patients. There is a possibility that AF patients show individual changes in mechanistic blood markers. Levels of specific blood markers may help in the selection of a drug therapy directed at this mechanism underlying AF. As such, we build on personalized medicine in AF.

## 7. Conclusions

As current treatments are not directed at the underlying pathophysiological processes driving AF, research is increasingly focused on the dissection of the molecular mechanisms. Mitochondrial dysfunction has been recently identified as a contributing factor in the electropathology of AF. However, current clinical diagnostic and treatment approaches for AF are not directed at the conservation of mitochondrial function and stabilization of the microtubule network. Key modulators of mitochondria may represent both novel druggable targets to attenuate AF progression and diagnostic targets to stage the severity of AF and predict new onset and recurrence after treatment. There is currently no complete agreement in the scientific community on the direct causal relationship between mitochondrial dysfunction and AF. This review paper summarizes the current state of research on this topic. Further investigation in this relevant area of research is encouraged. These findings may open a new field in the treatment and prevention of AF.

## Figures and Tables

**Figure 1 ijms-22-08463-f001:**
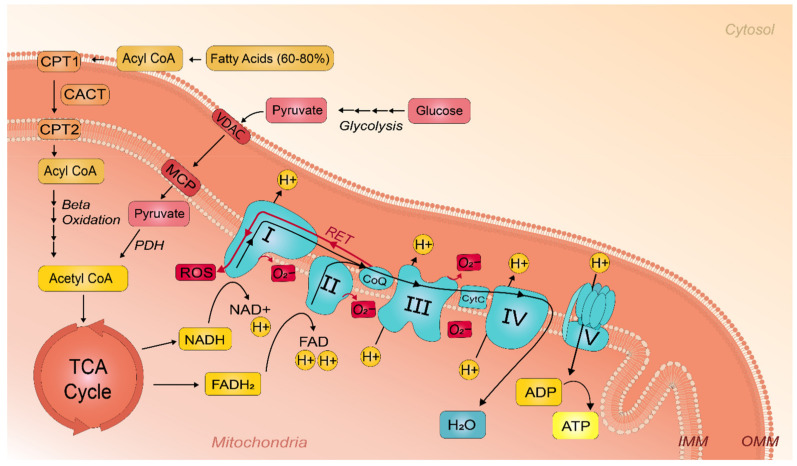
Schematic representation of ATP generation via fatty acid and glucose oxidation and leakage of reactive oxygen species in the mitochondria. Mitochondrial oxidative phosphorylation at the ETC is driven by the substrates NADH and FADH_2_, which originate predominantly from FA and carbohydrate oxidation. FAs pass the mitochondrial membrane as acyl-CoA by means of the carnitine acyltransferase membrane shuttles, consisting of CPT-1, CPT-2, and CACT. CPT1 converts acyl-CoA to acylcarnitine, which is subsequently transferred across the IMM by CACT. Inside the mitochondria, acylcarnitine is converted back into acyl-CoA by CPT-2. During β-oxidation, 2-carbon units are oxidized to produce acetyl-CoA, which is the substrate for the TCA cycle. Pyruvate traverses the mitochondrial membranes through VDAC and MPC. Once inside the mitochondrial matrix, pyruvate gets oxidized into acetyl-CoA to enter the TCA cycle. The reducing agents NADH and FADH_2_ are produced inside the TCA cycle. Electrons from NADH and FADH_2_ are donated to the complexes of the ETC, which translocate protons into the intermembrane space to establish a proton gradient for ATP synthesis. ROS are formed as by-products of the ETC. Not all of the electrons are transferred to the final electron acceptor as some leak out of the ETC into the mitochondrial matrix, followed by superoxide production. ROS overproduction contributes to oxidative stress and mitochondrial dysfunction.

**Figure 2 ijms-22-08463-f002:**
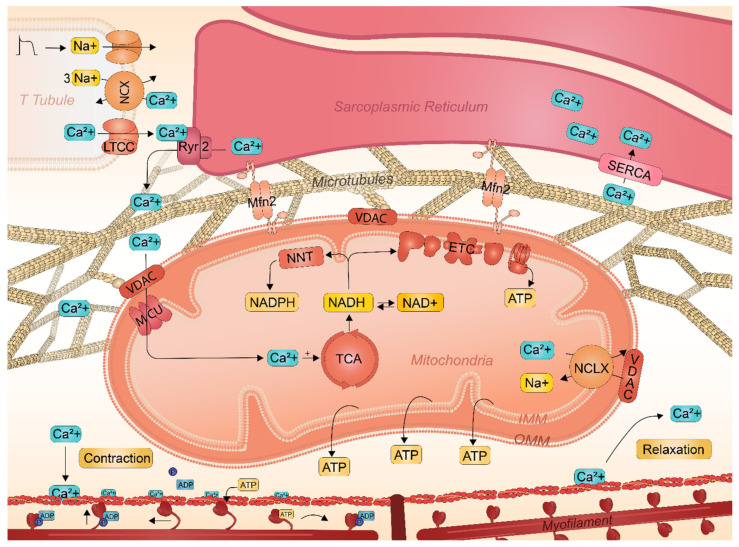
Schematic representation of Ca^2+^ transient handling and contraction in atrial cardiomyocytes and the role of SR and mitochondria. An action potential activates the Ca^2+^ channels in the T-tubule, resulting in an influx of Ca^2+^ into the cardiomyocytes. Subsequently, Ca^2+^ crosses the OMM through VDAC and is transported into the mitochondrial matrix via the MCU. Ca^2+^ efflux from the matrix is mediated by the NCLX. Cytosolic Ca^2+^ binds to cardiac troponin-C, which moves the troponin away from the actin binding site to form cross-bridges between actin and myosin. The myosin head binds to ATP and pulls the actin filaments towards the center of the sarcomere resulting in cardiomyocyte contraction. Dropping of the intracellular Ca^2+^ concentration returns the troponin complex to the resting position, thereby effectively inhibiting cardiac contraction and initiating relaxation.

**Figure 3 ijms-22-08463-f003:**
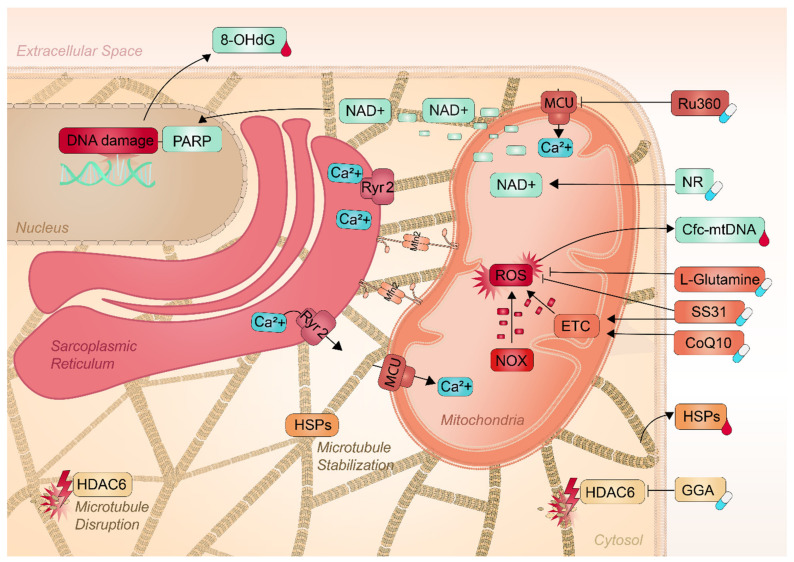
Schematic overview of the pathophysiological processes during AF and the potential biomarkers and therapeutic implications. Mitochondrial dysfunction has been identified as a contributing factor in AF. Modulators of mitochondrial dysfunction such as 8-OHdG and cfc-mtDNA may aid in the staging of the severity of AF, predicting postoperative AF (POAF) onset, and the outcome of AF treatments. In addition, various pharmaceutical- and nutraceutical compounds are under investigation nowadays. Pharmaceutical compounds such as SS31, Ru360, or HSP inducer GGA and nutraceutical compounds such as NR, L-glutamine, and CoQ10 may present novel therapeutic strategies to counteract AF-induced mitochondrial dysfunction.

**Table 1 ijms-22-08463-t001:** Tubular view of current studies on mitochondrial dysfunction and AF.

Author	Outcome
Emelyanova et al. (2016) [32]	AF is associated with a selective reduction in the mitochondrial ETC activity and increased oxidative stress in humans which may contribute to the progression of the substrate for AF.
Hu et al. (2019) [77]	HSP inducer GGA, reverses contractile and structural remodeling via restoration of the microtubule network in experimental AF.
Ozcan et al. (2019) [78]	The pathogenesis of AF is associated with energy deficit and metabolic dysregulation in human and mice atria due to mitochondrial dysfunction.
Wiersma et al. (2019) [23]	Mitochondrial dysfunction is involved in AF promotion, and compounds directed at the conservation of mitochondrial function protect against contractile dysfunction in Drosophila models for AF.
Xiao et al. (2010) [79]	Taxol, a microtubule stabilizer, prevents AF in an in vitro AF model using rabbit hearts. The microtubule stabilizer most likely prevents AF by reducing the level of ROS.
Xie et al. (2015) [22]	Mitochondrial-derived ROS oxidize atrial RyR2 in human cardiomyocytes. This leads to increased intracellular Ca^2+^ leak and impaired mitochondrial function, contributing to the pathogenesis of AF. Interestingly, reduced mitochondrial ROS production attenuates SR Ca^2+^ leak and prevents AF.
Zhang et al. (2014) [7]	Patients with AF show increased HDAC6 activity. Activation of HDAC6 induces the degradation of the microtubule network and contractile dysfunction in experimental and human AF. Drugs directed at the conservation of the microtubule network attenuate AF in a dog model for AF.

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
