# Peer review of "The Role of Mitochondrial Dysfunction in Atrial Fibrillation: Translation to Druggable Target and Biomarker Discovery"

_ijms, 2021, doi:10.3390/ijms22168463_

Round 1

Reviewer 1 Report

This review address the possible role of mitochondrial impairment in AF. The review is organized in several chapters describing the role of mitochondrial pathways in bioenergetics, including the different nutrients exploited by the heart tissue to produce electron donors, the transport of pyruvate and fatty acids within mitochondria, oxidative catabolism of both through the TCA cycle and beta oxidation, cell respiration through the respiratory chain and ATP synthesis by the oligomycin-sensitive ATP-synthase which exploits the proton electrochemical gradient across the inner mitochondrial membrane sustained by energy sustained by respiration. A further chapter is dedicated to the generation of ROS as a by-product of respiration, due to electron leakage (and also, but this is not mentioned by the Authors) by Reverse Electron Transfer (RET). AF is then described including the triggers and causes of this important arrhythmogenic cardiopathy. Oxidative stress caused by ROS production is claimed to contribute to the pathogenesis of AF. However, the role of ROS stress in AF is tenuous, there is no convincing experimental models for a specific or relevant role of ROS damage generated by mitochondria to determine impairment and malfunctioning of the a sino-atrial pacemaker, the formation of ectopic foci for the generation of the typical irregular arrhythmia that characterizes AF, atrial dilation which favours the onset of AF, etc. The only documented observation is an increase on 8-hydroxy 2-deoxyguanosine that has been reported in AF (in particular in the paper by Li J, Zhang D, Ramos KS, Baks L, Wiersma M, Lanters EAH, Bogers AJJC, de Groot NMS, Brundel BJJM. Blood-based 8-hydroxy-2'-deoxyguanosine level: A potential diagnostic biomarker for atrial fibrillation. Heart Rhythm. 2021 Feb;18(2):271-277. doi: 10.1016/j.hrthm.2020.09.017. Epub 2020 Oct 5. Erratum in: Heart Rhythm. 2021 May;18(5):845. PMID: 33031960. Notably, the increase of this biomarker is detected in blood, but its origin, whether from mtDNA or nuclear DNA, has not been investigated. The increase of mtDNA in circulating blood is a potentially interesting fin ding in AF but it could be due to extra-cardiac or extra-atrial damage due to the arrhythmia itself. Also, the generation of ROS is robustly circumvented in normal conditions by a powerful repertoire of anti-ROS enzymes, several of which are present inside mitochondria. Therefore, whilst it has clearly been proven that mitochondrial bioenergetics impairment can be associated with cardiac pump failure, the link between mitochondrial respiration or OXPHOS and AF is not convincing, and the review does not resolve this controversial issue. My advice is to re-write the review, in agreement with the Editors, discussing the mitochondrial role in heart failure, including as a yet unproven possibility a potential link between mitochondrial impairment and AF. The review as it is edited presently is not convincing, at least it did not convince me.  

Reviewer 2 Report

In this manuscript, Pool and colleagues present a review on the role of mitochondria in the pathogenesis of atrial fibrillation, with a proposed focus on therapeutical targets and potential biomarkers.

The topic is certainly interesting - there is plenty of clinical research on AF, but physiopathological mechanisms are still far to be completely elucidated. Furthermore, the authors have provided very beautiful and meaningful images, which enrich the quality of this piece.

Unfortunately, there are also several shortcomings.

In details:

- It is unclear what is the actual novelty of this review, compared to previous works (https://pubmed.ncbi.nlm.nih.gov/30179127/ ), some of which were actually very recently published and deal with the same topic (https://www.mdpi.com/2077-0383/10/11/2385). This is an important issue, since it seems that the novelty  may be reduced, and it is unclear how this review is different/novel from the others.
- Most of the part of the "Cardiac mitochondrial physiology" chapter are too loosely related to the topic presented in the title. In other words, if the review focus on the link between mitochondrial dysfunction and atrial fibrillation, a targeted background should be presented, rather than a very long introduction on the overall mitochondrial function/physiology. I strongly suggest the authors to revise the chapter, trying to 1) synthetize the information presented; 2) try to provide a more targeted review of the literature, according to the scope of their review.
- On the other side, the chapter on the link between mithocondrial dysfunction and the pathogenesis of AF (chapter 3) may be enriched with a more extensive presentation of the current evidence, or  with a tabular view of the most important studies on the field.
- Similarly, a tabular/graphical resumee of the chapter 4 and 5 would be very useful. I also suggest the authors to search if some currently used biomarkers/drugs may be used to diagnose/treat mitochondrial dysfunction in AF. This is important to give this review a clinical importance.
- What are the future perspectives on this arguments? Is there any ongoing study/research/trial on the topic? What are the grey zone that deserve the most attention when dealing with this topic? In which type of AF (for example, those related to infections?) does the mithocondryal pathology may be most important? All these questions may be addressed in a final chapter of this review since, from a clinical point of view, these are very important open issues and should be addressed more carefully.

Round 2

Reviewer 1 Report

The paper has been modified and this improved the literature contributing to validate the hypothesis of the Authors. Nevertheless, in several papers the occurrence of AF is associated with HF as well. In general, the detailed description of the physiology of mitochondrial bioenergetic metabolism and the production of ROS as a by-product of this fundamental process is correct but I maintain my opinion that there is no direct causative relation between mitochondrial dysfunction and onset or progression of AF. AF is often caused by severe structural/functional alteration of cardiac atrial and not infrequently it associated with heart failure. Mitochondria are essential organelles that provide substantial amount of ATP supply for the activity of the contractile structures of the heart and it is not surprisingly that mitochondrial abnormalities may occur secondarily to the profound lesions of atrial structure and tissue consequent to the several causes determining atrial fibrillation. Mitochondrial impairment may well be one of the many additional factors that contribute to the maintenance or worsening of AF, but there is no convincing experimental evidence for a direct role of mitochondrial dysfunction in determining the onset of the progression of AF, independently by the effects of mitochondrial impairment in heart failure, which may well be associated with AF. For instance, mitochondrial disorders are not associated with AF in the relevant literature available in PubMed, whereas there is a clear association between some mutations of either mtDNA or mitochondrion-related nuclear genes and heart failure. The ablation of MCU or of MCU regulator proteins such as MICU1 or MICU2 in mice are not associated with AF, despite the Calcium mitochondrial control is clearly impaired by each of these conditions. The observation that the modifications of guanosine residues are not reported to occur specifically in mtDNA is important since it makes this observation rather unspecific, and does not provide supprtive evidence in favour of a consistently reported mechanism by which oxidative damage of mtDNA nucleotides can determine further functional damage of mtDNA function and promote a vicious circle  Therefore, despite the laudable effort of the Authors, the evidence that mitochondrial dysfunction can determine or contribute to determine/worsen AF remains unconvincing, at least to me.

Reviewer 2 Report

The authors adequately addressed all the issues raised previously. I have no further comments.

Round 3

Reviewer 1 Report

The Authors dampened the conclusions, confirming that a relevant role of mitochondrial impairment in AF is controversial. I think that this is a matter of fact, and based on this acknowledgment the review can be considered a useful contribution on this issue.